# Micro-Raman for Local Strain Evaluation of GaN LEDs and Si Chips Assembled on Cu Substrates

**DOI:** 10.3390/mi15010025

**Published:** 2023-12-22

**Authors:** Enrico Brugnolotto, Claudia Mezzalira, Fosca Conti, Danilo Pedron, Raffaella Signorini

**Affiliations:** 1Department of Chemical Science, University of Padova, Via Marzolo 1, 35131 Padova, Italy; e.brugnolotto.1@research.gla.ac.uk (E.B.); claudia.mezzalira9@gmail.com (C.M.); danilo.pedron@unipd.it (D.P.); 2Consorzio Interuniversitario Nazionale per la Scienza e Tecnologia dei Materiali, Via G. Giusti 9, 50121 Firenze, Italy

**Keywords:** integrated circuits, assembly process, GaN, Si, thermomechanical local strain, mapping, Raman

## Abstract

Integrated circuits are created by interfacing different materials, semiconductors, and metals, which are appropriately deposited or grown on substrates and layers soldered together. Therefore, the characteristics of starting materials and process temperatures are of great importance, as they can induce residual strains in the final assembly. Identifying and quantifying strain becomes strategically important in optimizing processes to enhance the performance, duration, and reliability of final devices. This work analyzes the thermomechanical local strain of semiconductor materials used to realize LED modules for lighting applications. Gallium Nitride active layers grown on sapphire substrates and Si chips are assembled by soldering with eutectic AuSn on copper substrates and investigated by Raman spectroscopy in a temperature range of −50 to 180 °C. From the Raman mapping of many different samples, it is concluded that one of the leading causes of strain in the GaN layer can be attributed to the differences in the thermal expansion coefficient among the various materials and, above all, among the chip, interconnection material, and substrate. These differences are responsible for forces that slightly bend the chip, causing strain in the GaN layer, which is most compressed in the central region of the chip and slightly stretched in the outer areas.

## 1. Introduction

Light-emitting devices, or LEDs, are extensively studied in terms of emission efficiency, wavelength modulation, stability, and duration, as they are widely used for applications in various fields, such as interior and exterior lighting, computer, and smartphone screens, and for the automotive industry [1,2]. For applications in diverse environments, where they are often subjected to high temperatures and thermal cycling, one critical aspect that influences the widespread use of LEDs is their reliability and resistance, particularly in the face of extreme temperature conditions. Ensuring the reliability of LED devices under such challenging conditions has become a focal point in the pursuit of sustainable and efficient lighting solutions.

Gallium nitride (GaN)-based LEDs are particularly interesting for electronic applications in the visible range and for the realization of white emitting devices [3,4,5,6,7,8]. The development of microelectronic devices requires the use of materials with different chemical and physical characteristics that are interfaced with specific and optimized interconnections. The interconnections among various materials inevitably induce strains within the devices that affect their final performance [9,10,11]. The possibility of measuring and quantifying the degree of induced strain allows for the identification of assembly processes with lower impact, directing the processes toward the realization of better-performing devices.

Micro-Raman spectroscopy is a known technique for studying local mechanical strains in microelectronic devices and structures. It has the advantages of being fast, non-contact, and non-destructive while simultaneously allowing for micrometer spatial resolution [12,13,14]. The Raman signal can be collected with micrometric resolution at different positions in a sample to achieve a complete evaluation of the strain distribution. Alternative techniques for characterizing residual strains in thin films present several drawbacks. The substrate curvature method is mainly based on quantifying the change in substrate curvature before and after the film deposition. It provides a strain value averaged over the film area. Due to its approximate strain distribution algorithms, it is not suitable for residual strain measurements of thick films or for films with an asymmetric strain distribution. Furthermore, it does not provide information on the local strain. X-ray diffraction (XRD) examines residual strains at different depths of the film structure due to the penetration capacity of the X-ray, but it does not satisfy the requirement for mechanical measurement in microelectronic devices due to its inherent limitations in spatial resolution and accuracy [15]. Cross-sectional transmission electron microscopy (XTEM) [16] and convergent beam electron diffraction (CBED) [17] can provide information on strain on a very high resolution scale (nm), but destructive sample preparation is required, causing strain relaxation in specific directions; additionally, the interpretation of XTEM images requires extensive modeling. Moreover, TEM techniques are very time consuming and cannot be integrated into production lines, unlike micro-Raman measurements, which have a high potential for in-line metrology applications in the semiconductor industry [18]. Due to these considerations, the Micro-Raman technique may be the most suitable for this purpose. The principle behind the correlation of the Raman bands and the local strain is found considering that a mechanical or thermal strain affects the frequencies of the optical phonons through a displacement of the atoms involved in their production from their initial equilibrium positions, induced by a thermal expansion coefficient mismatch of the different materials.

This paper analyses the strain induced in the GaN emissive layer by the assembly process as a function of assembly parameters such as the support material and applied pressure. In parallel, the investigation of Si chips, which are used as substrates for the deposition of GaN LEDs, allows one to obtain additional information on the origin of strain development.

The developed strains were characterized in commercial Si chips and LEDs, produced using UX: 3 chip technology by OSRAM Opto Semiconductors GmbH [19,20]. Using soldering technology, LEDs and Si chips were mounted on copper plates with a surface area of 1 cm^2^. The interconnection layers comprised a tin–gold eutectic alloy (80/20 Au/Sn by weight). The mounting parameters, varied during assembly, are the applied soldering pressure (from 0 to 21 N) and the thickness of the copper substrate (from 0.2 to 1.5 mm).

The assemblies were studied using Raman spectroscopy, a non-destructive technique that gives, as one of its advantages, the possibility of analyzing the same sample before and after the soldering process and the possibility of obtaining strain maps using a micro-Raman setup. LEDs and Si chips were characterized with a micro-Raman using the excitation wavelength of an Argon laser at 514 nm. Raman mapping measurements were performed to determine the difference in strain induced at various positions of the sample at three different temperatures, 180, 20, and −50 °C, to investigate the effect of thermal expansion on strain in the layers.

From the recorded spectra, the variation in the position of the $E_{2}^{H}$ Raman peak of GaN and the LTO Raman mode of Si allow the calculation of the local strain present in the active layer, from which the strain induced by the assembly process can be quantified.

The results showed that the strain in the GaN layer is mainly attributable to differences in the thermal expansion coefficient between the various materials and, above all, among the chip, the interconnect material, and the substrate. This appears to create forces that bend the chip, causing strain in the GaN layer, which is most compressed in the core region of the chip and slightly stretched in the outer regions. Samples soldered on thin copper plates show lower strain than those on thick plates. Thin chips favor relaxation of the strain of the chip due to their deformation capacity, whereas thick chips, which are less flexible, restrict only the strain to the chip.

Since the investigated GaN LEDs are also composed of a thin layer of Si, from the comparison between the GaN sample and the Si-only sample, it was possible to distinguish the induced strain of the GaN layer itself from that of the overall device.

## 2. Materials and Methods

### 2.1. Si and GaN Samples

Silicon chips (001) oriented with (1500 × 1500 × 120) μm^3^ dimensions (sample Si1–Si3) have been soldered on a copper substrate of 5 × 5 mm size dimensions and 1 mm thickness. A thin layer of Au80Sn20 eutectic alloy with a thickness of 25 μm was used as an interconnecting component. The experimental conditions are widely described in [21,22,23]. The structure of the assembled Si chip is shown in Figure 1a, and the image of the final sample is shown in Figure 1c.

This work investigated blue LEDs purchased from the Light Avenue distributor. Chips are typically 1000 × 1185 µm in size and 120 µm in height; they were produced using the UX:3 chip technology developed by OSRAM Opto Semiconductors GmbH, Regensburg, Germany [19,20]. Three LEDs (SV1–SV3 LED) were characterized in their pristine state before the assembly procedure. Using an eutectic layer of gold tin (Au80Sn20), with a thickness of 25 μm, four LEDs were soldered on a copper substrate of approximately 6 × 6 cm in size and 1.5 mm thickness (samples S1 and S3) or 0.2 mm thickness (samples S2 and S4). Additional details on LED configuration and soldering experimental parameters, such as atmosphere and temperature profiles and the bonding time used for the assembly process, are given in [24,25].

The soldering process was carried out under two conditions: S1 and S3 without pressure, and S2 and S4 with 21 N pressure. The assembled LED scheme is reported in Figure 1b, and the sample photo is shown in Figure 1d. Table 1 shows a complete list of the investigated samples and their main characteristics.

### 2.2. Set up of Raman Measurements

Raman measurements were performed on a Micro-Raman setup. A single-line argon ion laser was used as an excitation light source, with two principal lines at 488 and 514.5 nm (Spectra Physics Stabilite 2017 output power 1 W). The 488 nm radiation was filtered out, and a half-wave plate was used to control the polarization of the incident light. Optical density filters were arranged on a remotely controlled reel to regulate the light intensity hitting the sample. The laser beam is coupled to a microscope (Olympus BX 40, Tokyo, Japan) and focused on the sample using a 20× objective (Olympus SLMPL, NA D 0:75, Tokyo, Japan); the typical spot diameter in the focus is 3 μm. The back-scattered Raman signal is separated from the Rayleigh scattering by an edge filter and analyzed with a 320 mm focal length imaging spectrograph (TRIAX-320 ISA, Horiba, Lyon, France) and a liquid nitrogen-cooled CCD camera (Spectrum One, JobinYvon, Horiba, Lyon, France). During analysis, a temperature-controlled stage (Linkam THMS600, Tadworth, UK) was used to cool and heat the samples and keep the system at the desired temperature.

The Raman signal of the samples was measured by mapping the surface of the Si chip (see Figure 1e,f) in 10 × 10 positions and the surface of the LED in 20 × 20 positions at −50, 20, and 180 °C. Each point was recorded using the 20× microscope objective, and the spectrograph slit was set at 40 μm. The spectra were obtained by averaging five repeated measurements with an acquisition time of 5 s for each spectrum (5 s × 5 times).

### 2.3. Determination of Strain Induced from Raman Measurements

When stress, defined as a force per unit of area, is applied to a material, it induces strain on the material itself [26]. From a microscopic point of view, the positions of atoms or the chemical bonds within the material change compared to their equilibrium states. This, in turn, shifts the Raman peak position to higher (for compressive forces) or lower (for tensile forces) frequencies. Therefore, the magnitude and direction of the Raman peak shift are related to the applied stress and the corresponding induced strain. They are strictly related to the characteristics of the investigated material, such as the elastic compliance constant (S_ijkl_) and the specific crystallographic axes.

The theory of deformation potentials describes the change in interatomic potentials as a result of atomic displacement and strain. The linear deformation potential theory can be used to mathematically describe the effect of strain on the energy of the phonon modes; this consists essentially of a first-order truncated Tailor expansion of energy in powers of a parameter representing lattice strain, hence the ‘linear’ part of the name. This approximation remains valid for sufficiently low strains.

An application of the deformation potential theory to the case of cadmium sulfide (CdS) was explored by Briggs and Ramdas [27]. The calculation was based on group theory, resulting in general expressions linking the energy (eigenvalue) shift (∆E) of a phonon eigenstate, corresponding to a vibrational mode, of a P_6mc_ crystal to its strain (ϵ).
$$∆E=a\left(\epsilon_{xx}+\epsilon_{yy}\right)+b\epsilon_{zz}\pm c\sqrt{{\left(\epsilon_{xx}-\epsilon_{yy}\right)}^{2}+4{\epsilon_{xy}}^{2}}$$
Constants *a*, *b*, and *c* are phonon deformation potentials and vary by material. Taking into account (i) the conversion from energy to wavenumber, (ii) the elastic strain relation $\underline{\epsilon }=s̿\underline{\sigma }$ where s̿ is the elastic compliance matrix, (iii) the case of symmetric in plane strain $\epsilon_{xx}=\epsilon_{yy}$, and (iv) negligible shear strain in the *xy* plane $\epsilon_{xy}=0$, the equation can be rewritten in a simplified form as
$$∆\omega =a^{\prime }\sigma_{c-plane}+b^{\prime }\sigma_{zz}$$

If the strain along the *c*-axis is negligible, a final expression can be derived as follows:$$∆\omega =-K\sigma_{c-plane}$$
where the negative sign means that a higher Raman shift value results from compressive strain. Some values of *K* values reported in the literature are summarized in Table 2 for the case of GaN and allow for a quick and straightforward conversion from Raman shift to strain. Various methods have been employed over the years to determine the value of *K*, so today, both experimental and calculated values are available, and experimental data also confirm the linear approximation.

For GaN crystals in wurtzite symmetry, the phonon shifts (Δω) of the $E_{2}^{H}$ Raman mode are represented by the following simplified equation, which allows the calculation of the induced strain of the experimental Raman shift [22]:$$∆\omega (E_{2}^{H})=-1.55\cdot 10^{-9}\left(\sigma_{xx}+\sigma_{yy}\right)$$
where $\left(\sigma_{xx}+\sigma_{yy}\right)$ is the average value of the total induced “in-plane strain”, i.e., the sum of the inverse piezoelectric (IPE) and thermoelastic (TE), strains in the *x*–*y* plane with strain components $\sigma_{xx}$ and $\sigma_{yy}$. The calculated constant $K=-1.55\cdot 10^{-9}$ is comparable to experimental and computed data reported by Choi et al. [36] and Bagnall et al. [37], respectively.

Silicon is a semiconductor material with the diamond symmetry $O_{h}^{7}$, consisting of two interpenetrating face-centered cubic structures, displaced along the body diagonal to the cubic cell by one-quarter the diagonal length. The method used to quantify the local residual stress of a silicon sample is similar to that used for GaN. Based on the Raman spectrum, it involves the resolution of a secular matrix containing the phonon deformation potentials and the components of the strain tensor along each direction [13,38]. Considering the components of the strain tensor related to the strain tensor (*σ*) through the elastic compliance tensor elements of silicon (reported in Table 3), the biaxial stress along the [100] and [010] directions (*xy* plane), for backscattering from a surface (001), is given by the following equation [39]:$$\left(\sigma_{xx}+\sigma_{yy}\right)=-2.26\cdot 10^{5}\frac{{\Delta \omega }_{j}}{\omega_{0}}$$

## 3. Results

### 3.1. Raman Spectra of Si and GaN Samples

The Raman spectra of the Si sample, recorded at −50, 20, and 180 °C, are reported in Figure 2a. The spectra present an intense Raman signal, centered at ~519 cm^−1^ at 20 °C, associated with two transverse and one longitudinal branch, an LTO branch, and a small broad peak at 950 cm^−1^, corresponding to the second-order Raman signal. These signals are comparable to those reported in the literature [42]. The peak is shifted to 521 cm^−1^ at −50 °C; at 180 °C, the peak is shifted to 515.5 cm^−1^. The peaks at 116 (not reported in the figure) and 265.5 cm^−1^ are plasma lines used to normalize the spectra in wavelength.

Gallium Nitride is a III–V semiconductor, possessing a stable wurtzite crystalline structure. First-order Raman scattering is allowed at the center zone point (Γ). The Raman modes $A_{1}$, $E_{1}$, $E_{2}$, are clearly visible in the Raman spectra reported in Figure 2b: $E_{2}^{L}$ centered at 144 cm^−1^, $E_{2}^{H}$ centered at 568 cm^−1^, and an *LO* phonon peak, $A_{1}(LO)$ centered at 736 cm^−1^, and a *TO* phonon peak, $A_{1}(TO)$ centered at 560 cm^−1^, while $E_{1}$ can be observed as a small shoulder of the $E_{2}^{H}$ peak. These are typical phonon spectra of a hexagonal GaN layer grown on a sapphire (1000) substrate [43]. Furthermore, with GaN, small changes to higher frequencies at −50 °C and lower frequencies at 180 °C are visible as a function of temperature.

The positions of the GaN $E_{2}^{H}$ and Si LTO phonon modes are strictly related to temperature, strain, and electric field and indicate the presence of local lattice strains induced during the assembly process. From the Raman mapping of the samples, the presence of locally induced strains will provide information on the quality of the assembly.

### 3.2. Samples at Room Temperature

The Si and LED samples are characterized by a significant variation in the peak position of the LO and $E_{2}^{H}$ throughout the sample, with the highest variation along the diagonals of the samples. For example, the position of the $E_{2}^{H}$ Raman peak, of S1–S4 LED samples, collected at 20 °C along the diagonal, is reported in Figure 3a–d.

Raman data show the difference between soldering with and without pressure. Sample S1, assembled without pressure on a 1.5 mm thick copper substrate, shown in Figure 3a, presents a greater peak shift (${∆\omega }_{S1}=\omega_{max}-\omega_{min}=2.34$ cm^−^^1^), compared to the other samples, Figure 3b–d assembled on a thinner copper substrate or with pressure (${∆\omega }_{S2-S4}=1.64÷1.86$ cm^−^^1^). The use of pressure or thinner substrates is reflected in a lower strain on the final assemblies.

Table 4 reports all the measured values of the GaN and Si samples before and after assembly. The unassembled samples show a minimal variation in frequency position, 0.22 cm^−^^1^ for Si and 0.46 cm^−^^1^ for GaN, while the Si chip shows ${∆\omega }_{Si}=1.88$ cm^−^^1^, a value in the middle in between those observed for S1 and S2–S4 LED samples.

### 3.3. Raman Mapping of Samples at Different Temperatures

The Si chip and GaN LED samples, unmounted (Si and SV1 samples) and assembled (A_Si and S1 samples), were thoroughly investigated by collecting the Raman spectra on the entire surface (reported panels e and f of Figure 1), at different temperatures: −50, 20, and 180 °C. The values of the Raman shift of the LO peak of Si and $E_{2}^{H}$ peak of GaN Raman shifts, collected on the samples, are reported in Figure 4 for Si (panels a, c, e) and A_Si (panels b, d, f) samples and in Figure 5 for SV1 (panels a, c, e) and S1 (panels b, d, f) samples. These figures show how the temperature affects the Raman shift value of all samples. As the temperature increases, the Raman shift values decrease: the LO Raman mode of Si is in the range of 520.6–523.2 cm^−1^ at −50 °C, 518.6–520.5 cm^−1^ at 20 °C, and 515.3–515.9 cm^−1^ at 180 °C, while the $E_{2}^{H}$ Raman mode of GaN is in the ranges 567.5–570.3 cm^−1^ at −50 °C, 566.9–569.1 cm^−1^ at 20 °C, and 565.1–565.9 cm^−1^ at 180 °C.

The maps of the assembled chip reveal a symmetric distribution over the area of the samples, with higher values in the center at all temperatures. In addition, the center region is the most stressed area and is more affected by thermal processes than the corners. This distribution is not visible in the pristine samples, which are almost uniform at all temperatures.

Similar considerations can also be made for the GaN LED, with minor differences. Figure 5b,d,f highlights how the Raman shift of the assembled GaN LED is not perfectly symmetric from left to right. This asymmetry is due to the morphology of the chip. In particular, the off-center position of the GaN dye compared to the total area of the LED, for which one of the sides is occupied by the contact strip p, as reported in Figure 1c,e. The results indicate that the induced strain may also be due to bending of the LED chip. Furthermore, the strain distribution in the unassembled samples is always mainly uniform at all temperatures. Still, it settles on low Raman shift values at low temperatures (−50 and 20 °C) and on high values at 180 °C.

Finally, it is possible to outline that the Raman shift of both Si and GaN samples, measured at room temperature before and after the thermal treatments, is unchanged.

### 3.4. Strain Evaluation

To calculate the strain for the Si and GaN layers, a reference value is necessary: the best possible reference would be a layer of strain-free Si and GaN, respectively; it can be found by taking either a literature value or an internal value from measurements, such as the unmounted samples. For these considerations, the Raman frequency of the unmounted sample was assumed as a reference value to evaluate the strain values of the assembled samples. Therefore, positive and negative values were observed; positive strain values correspond to tensile strain, while negative strain values correspond to compressive strain.

The strain distribution of the different samples was determined from the Raman mapping data; color maps in Figure 6 report the strain values of the Si and GaN samples, determined on the entire surface of the samples at 20 °C. The maps have an almost symmetric distribution with tensile strains at the corners and compressive strains at the central part of the samples.

Data calculated at the different set temperatures are reported in Table 5 for the Si chip and Table 6 for the GaN LED, where the maximum and minimum stress values are reported, together with the average values and the maximum variation along the sample. At 20 °C, Si shows an evident prevalence of contractile deformation with a minimum value slightly higher than that obtained for GaN. At the same temperature, GaN shows both tensile and compressive stress values, with high intensity in both directions and comparable to each other. Tensile stresses are recorded in the peripheral areas of the device, and compressive stresses are recorded in the central area.

At low temperatures (−50 °C), a minor tensile stress appears, and the tensile stress increases by the same amount in the Si sample, while GaN shows an increase in tensile stress. At 180 °C, GaN only shows tensile stress, which, in absolute value, significantly exceeds the stresses that affect Si at 180 °C, while Si still exhibits a lower contractile stress.

The results presented in Table 5 and Table 6 suggested that at 20 °C and −50 °C, Si is much more affected by the tensile strain with a negligible contribution of the compressive strain. However, GaN is affected by both types of strain approximately equally. However, at 180 °C, the stronger strain load is on GaN rather than Si. By comparing the average strain values of the two samples, it appears that the Si sample always exhibits compressive strain at all temperatures. At the same time, the GaN values range from positive values at low temperatures to negative values at high temperatures.

The summary of these results is graphically reported in Figure 7a,b, where the minimum (Min) and maximum (Max) strain values are reported, together with the maximum strain variation in the samples (Delta) calculated as the difference between the maximum and the minimum strain value. In the figure, the Delta value obtained is reported above the maximum strain value measured. The samples show a similar behavior at increasing temperatures; in particular, the GaN sample always shows values higher than those of Si.

## 4. Discussion

Raman spectroscopy allows analysis of the local strain in assembled devices. From the Raman mapping of different samples, the strain distribution on the samples can be qualitatively observed and quantitatively determined, and the strain type can be characterized as tensile or contractile. Sample S1, consisting of a GaN LED soldered on a 1.5 mm thick copper substrate under pressureless conditions using a gold tin interconnect material, shows the largest strain variation among the analyzed samples. At room temperature, tensile stresses of 548 ± 22 MPa are recorded in the peripheral areas of the device. In comparison, compressive stresses of −716 ± 22 MPa are recorded in the central area, giving a maximum stress variation of 1264 ± 44 MPa in the sample. High compressive stress values are common for all gold–tin interconnected samples using the 1.5 mm copper substrate, and similar values are also found with the A_Si sample. On the contrary, lower values were observed for GaN LEDs mounted on a thinner substrate. These observations suggest that part of the stress is transferred to the substrate and generates a bending of the substrate. Consequently, a bending of the chip is produced that is no longer slightly concave but assumes a more wavelike shape.

The variation of strain, observed as the temperature varies, can be understood by comparing the temperature of the sample with that of the assembly, considering the thermal mismatch due to the different thermal expansion coefficients of the materials involved. Data from the corresponding literature are reported in Table 7: copper has a high thermal expansion coefficient (15–20 ppm/°C) compared to silicon (around 2.4 ppm/°C) and GaN (around 4 ppm/°C).

The results at different temperatures clearly show that the strain variation in the GaN layer is characterized by much lower values when measured at 180 °C. On the contrary, the highest strain variations were recorded at −50 °C. A possible explanation is suggested by considering the sample assembly processes: for soldered samples, the temperature is mainly between 220 °C and 250 °C. This means that liquid solder connects an expanded chip to an expanded substrate at these temperatures. As the cooling ramp begins, all materials shrink according to their thermal expansion coefficient, which differs between the chip components, between the chip and the interconnect, and between the interconnect and the substrate. Even if the existence of a temperature range for the solder with a still soft interconnect that can plastically adapt to the shrinking sizes of the sample components is assumed, mechanical rigidity is achieved, and elastic strain is built into the chip.

At 180 °C, where the temperatures are very close to the assembly process temperature, the dimensions of the sample’s components are closer to the size they had during the soldering process; therefore, most of the elastic strain is relaxed, resulting in the lower strain values registered. On the opposite side of the range, the −50 °C experiment represents a situation in which the sample components have a very different size than the one they had during the assembly process and, therefore, the resulting measured strains are higher.

The assembled LED will likely be slightly curved due to the forces built in due to thermal expansion coefficient mismatch. This is consistent with the strain results obtained, which can be interpreted as a slight concave curving of the GaN surface.

The comparison between the Si sample and the GaN LED, reported in Figure 8, allows us to make some interesting final considerations. The thicknesses and number of interfaced layers influence their deformation. The Si sample represents the substrate on which the active GaN layer is deposited. From the comparison between the maximum stress values developed within the two assemblies, it is possible to observe that both, at low and room temperature, present a difference equal to approximately 450 MPa, which becomes approximately 50 MPa at 180 °C. This fact indicates that the main component of stress is attributable to the Si substrate, which is thicker than that of GaN.

It has already been demonstrated that most LED products with high local strain have poor photoelectric performance and decrease in a short time [49], and in addition, the reliability also mainly includes the lifetime [50]. Recently, many articles have mentioned that the most important factor in affecting reliability is the thermal management problem of the LED packaging structure [51,52]. Therefore, the reliability of LED products depends on the cooling channels with reasonable design and on the materials with high heat conduction performance. Moreover, the local strain in the chip affects the emission characteristics of LEDs, the operational stability is reduced, and the strained interfaces cause an increased defect density. Power conversion efficiency can be strongly affected by local strain. In the current study, the distribution of strain inside the device has been systematically investigated by testing the effect of different assembly parameters and the temperature dependence of the Raman signal. Therefore, the characteristics of the substrates and assembly parameters that induce the least stress development have been identified in this work, from a comparison between the various samples.

One of the particularly interesting results to underline is related to the position of the Si and Gan Raman peaks recorded before and after thermal treatments: since they remain unchanged, it indicates reversibility of the process and, therefore, a good thermal resistance of the samples. This is a good result, since many electronic devices have to operate at high temperatures and are exposed to temperature cycles with large temperature differences; thus, they have to withstand strong cyclic thermo-mechanical stress. Several failure modes are possible up to intense cracking of the solder joints. Further advanced testing is required to verify the formation of defects in welded components and interconnection materials when a large number of thermal cycles occur. As an example, crack propagation in solder joints has already been identified by measuring the relative thermal resistance using transient thermal analysis [53,54].

Therefore, we can conclude that the choice of the type of material used as a substrate for the assembly of the samples and the determination of its thickness, together with the definition of the experimental parameters used in the assembly phase, are vital in determining the strains developed. Furthermore, the possibility of using micro-Raman for process control is beneficial for optimizing assembly processes.

## Figures and Tables

**Figure 1 micromachines-15-00025-f001:**
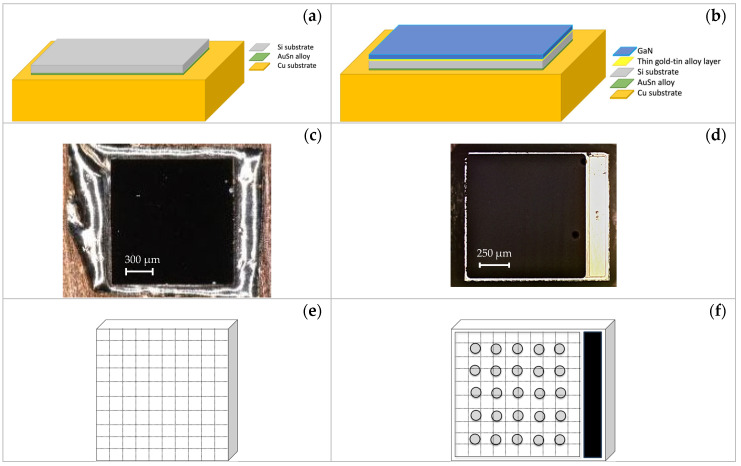
Schemes of assembly and photographs of the tops of the Si chip (**a**,**c**) and LED (**b**,**d**). Top view scheme of the Si (**e**) and LED (**f**) samples with schematic subdivision of the surface sample used to map the surface.

**Figure 2 micromachines-15-00025-f002:**
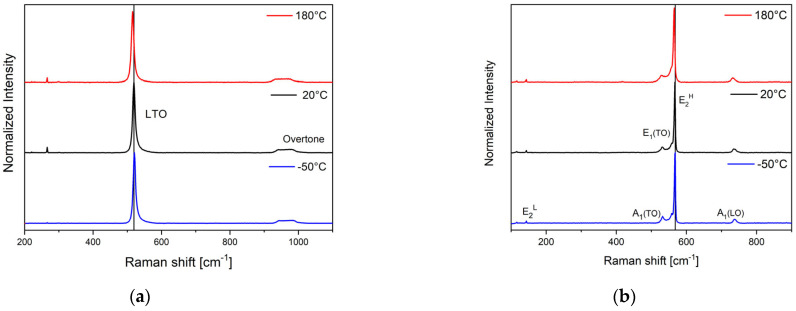
Raman spectra of silicon (**a**) and GaN (**b**).

**Figure 3 micromachines-15-00025-f003:**
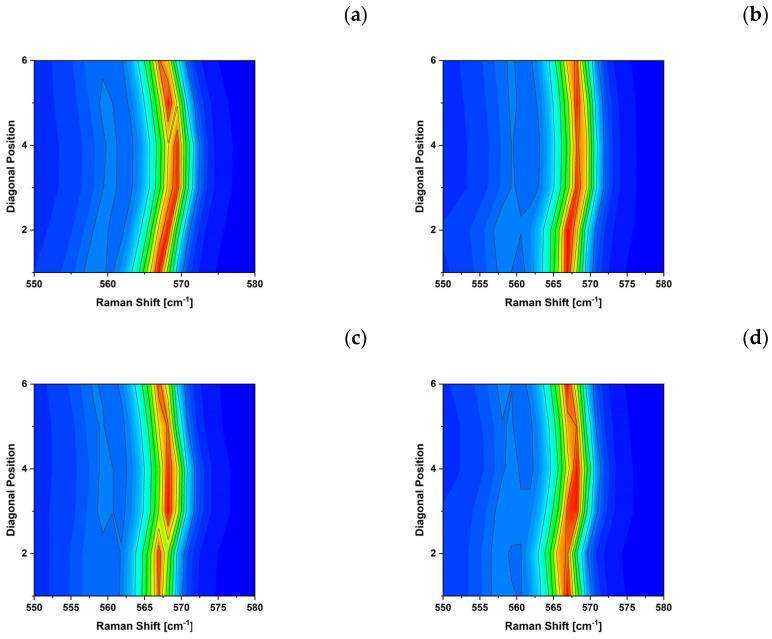
Raman shifts of the $E_{2}^{H}$ peak of the LED samples S1 (**a**), S2 (**b**), S3 (**c**), and S4 (**d**), collected at 20 °C along the diagonal of the samples. The colors indicate the Raman Intensity ranging from 0 (Blue) to 3 × 10^4^ (Red) Counts.

**Figure 4 micromachines-15-00025-f004:**
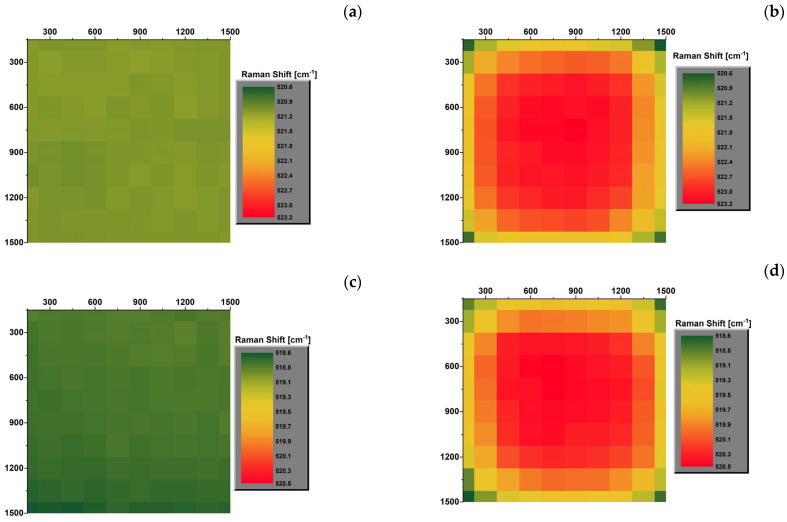
Mapping of the Raman shift of the LO peak of silicon in pristine unmounted (**a**,**c**,**e**) and assembled (**b**,**e**,**f**) Si chip at −50 (**a**,**b**), 20 (**c**,**d**), and 180 °C (**e**,**f**).

**Figure 5 micromachines-15-00025-f005:**
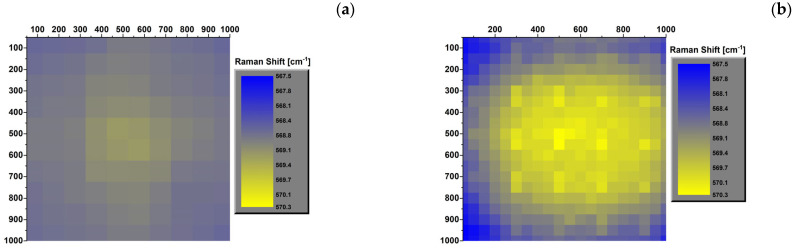
Map of the Raman shift of the $E_{2}^{H}$ peak of GaN in the VS1 and S1 assemblies at the temperatures of −50 (**a**,**b**), 20 (**c**,**d**), and 180 °C (**e**,**f**).

**Figure 6 micromachines-15-00025-f006:**
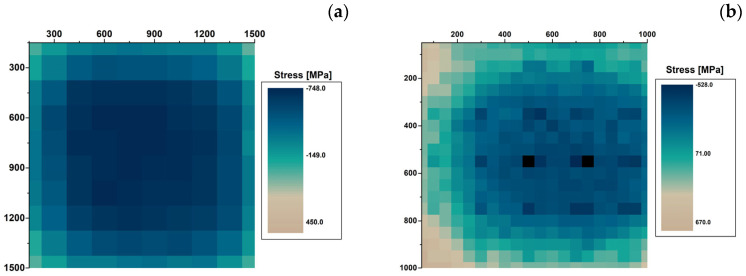
Stress maps of Si (**a**) and GaN (**b**) at 20 °C.

**Figure 7 micromachines-15-00025-f007:**
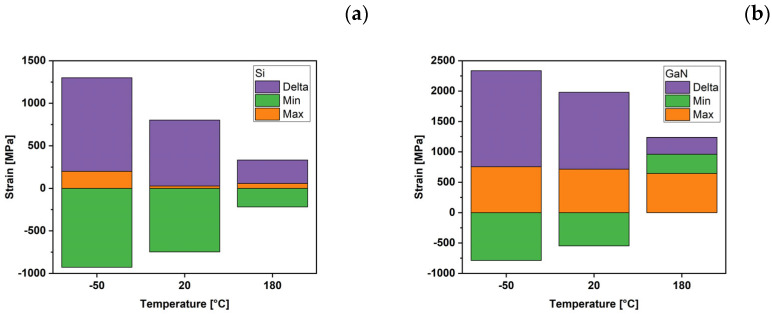
Maximum (Max) and minimum (Min) strains developed in Si (**a**) and GaN (**b**) samples, and differences (Delta) between them as a function of the temperature.

**Figure 8 micromachines-15-00025-f008:**
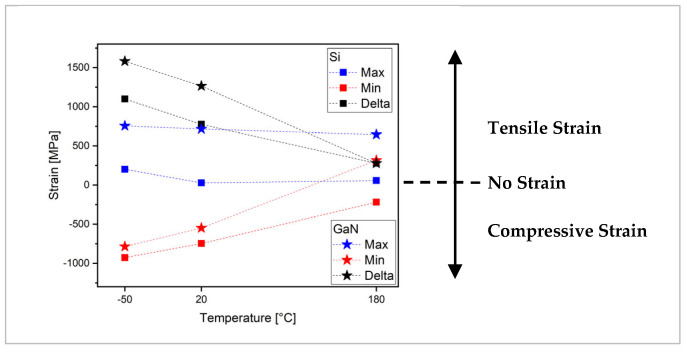
Strain developed on Si (squares) and GaN (stars) samples as a function of the temperature. Red and blue symbols refer to the minimum and maximum found values, respectively. Black symbols refer to the total strain developed (Delta).

**Table 1 micromachines-15-00025-t001:** Main characteristics of the investigated Si and GaN samples.

Sample Name	Chip Dimensions [mm^3^]	Number of Investigated Samples	Cu Substrate Dimensions [mm^3^]	Pressure [N]
Si1–Si3	1.5 × 1.5 × 0.12	3	/	/
A_Si_p		3	5 × 5 × 1	5
A_Si		3	5 × 5 × 1	/
SV1-3 LED	1 × 1.2 × 0.12	3	/	/
S1		1	6 × 6 × 1.5	/
S2		1	6 × 6 × 0.2	/
S3		1	6 × 6 × 1.5	21
S4		1	6 × 6 × 0.2	21

**Table 2 micromachines-15-00025-t002:** *K*-values evaluated in GaN samples grown on different substrates.

*K* [cm^−1^ GPa^−1^]	Growth Process	Substrate Type	Thickness [µm]	References
2.7	MOCVD (samples) CHVPE (reference)	6H Silicon carbide	0.5–3 300	[28]
2.9	MOPVE	Sapphire	unspecified	[29]
3.93	HVPE	Spinel	10–100	[30]
4.2	MOCVD, MBE	Sapphire (AlN buffer, homoepitaxial buffer)	0.5–5	[31]
4.3	MOCVD	Si, SiC	1	[32]
4.86	MBE	Sapphire (different orientations, GaN buffer)	0.3–1	[33]
6.2	MOCVD	Sapphire (50 nm AlN buffer)	2.5–50	[34]
7.9	MOCVD	Sapphire (20–250 nm AlN buffer)	0.75	[35]

**Table 3 micromachines-15-00025-t003:** Elastic compliance constants and phonon deformation potential values for silicon [35,40,41].

$S_{11}[{Pa}^{-1}]$	$S_{12}[{Pa}^{-1}]$	p $\left[s^{-2}\right]$	q $\left[s^{-2}\right]$
$7.68\cdot 10^{-12}$	$-2.14\cdot 10^{-12}$	$-1.85\cdot {\omega_{0}}^{2}$	$-2.31\cdot {\omega_{0}}^{2}$

**Table 4 micromachines-15-00025-t004:** Raman peak shift along the diagonal of the Si and GaN samples.

Sample	${∆\omega }_{Max-Min}$ [cm^−1^]		
	−50 °C	20 °C	180 °C
Si	0.11	0.22	0.23
A_Si_p	-	-	-
A_Si	2.59	1.88	0.55
SV1-3 Pristine LED	0.56	0.46	0.53
S1	2.81	2.34	0.82
S2	0.65	1.64	0.39
S3	2.31	1.86	0.52
S4	1.87	1.65	0.87

**Table 5 micromachines-15-00025-t005:** A_AuSn sample stress values [MPa].

Temperature [°C]	Maximum Value	Minimum Value	Average Value	Maximum Variation
−50	200 ± 22	−928 ± 22	−535 ± 30	1100 ± 44
20	28 ± 22	−747 ± 22	−471 ± 21	775 ± 44
180	57 ± 22	−219 ± 22	−110 ± 6	276 ± 44

**Table 6 micromachines-15-00025-t006:** GaN sample stress values [MPa].

Temperature [°C]	Maximum Value	Minimum Value	Average Value	Maximum Variation
−50	755 ± 22	−787 ± 22	−200 ± 35	1542 ± 44
20	716 ± 22	−548 ± 22	−35 ± 27	1264 ± 44
180	645 ± 22	316 ± 22	515 ± 6	329 ± 44

**Table 7 micromachines-15-00025-t007:** Properties of the materials of the assembly components.

Material	Thermal Expansion Coefficient [ppm/°C]	Elastic Modulus [GPa]	References
Silicon	2.3–2.6	191	[44]
Gold	14.3	61	[45]
Pure Copper	15–20	131	[17]
Al_2_O_3_	8.1	330	[18]
Au80Sn20 Alloy	16	68	[46,47]
GaN	4.0	Anisotropic elasticity	[16,48]

## Data Availability

Data are contained within the article.

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
