# Peer review of "Micro-Raman for Local Strain Evaluation of GaN LEDs and Si Chips Assembled on Cu Substrates"

_micromachines, 2023, doi:10.3390/mi15010025_

Round 1

Reviewer 1 Report

Comments and Suggestions for Authors

The manuscript presents a detailed analysis of the local strain in GaN LEDs and Si chips using Micro-Raman spectroscopy, providing valuable insights into the strains induced during assembly processes. The manuscript provides valuable insights into the field of semiconductor material analysis. The methodology is solid and valuable, and the data analysis is thorough. The manuscript could be accepted with minor revisions to enhance its overall impact and clarity.

1. The manuscript provides robust data analysis but lacks sufficient discussion on the broader implications and significance of the findings. It would be beneficial to expand on how this research contributes to the existing body of knowledge and its potential applications in the field.

2. The manuscript presents a detailed temperature-dependent strain analysis of the semiconductor materials. While the results at different temperatures provide valuable insights, it is suggested that the authors expand on the implications of these temperature variations for practical applications. How do these temperature-induced strains impact the overall performance and reliability of the semiconductor devices in real-world scenarios? Additionally, a discussion on potential methods to mitigate or manage these temperature-induced strains in practical applications would greatly enhance the applicability of the research findings.

3. Figure 1 (panels c and d): The microscopic images lack scale bars and size annotations. Adding these details is crucial for understanding the scale of the observed phenomena.

4. Figures 4 and 5: The text in these figures is too small and challenging to read. Consider resizing or redesigning these figures for better readability.

Comments on the Quality of English Language

The manuscript requires language editing to improve readability and professionalism. The current level of English distracts from the content's quality. I recommend a thorough language revision, preferably by a native English speaker with technical knowledge in this field.

Author Response

We thank the referee very much for the time he/she dedicated to read our paper and for the valuable comments provided.

  1. The manuscript provides robust data analysis but lacks sufficient discussion on the broader implications and significance of the findings. It would be beneficial to expand on how this research contributes to the existing body of knowledge and its potential applications in the field.

 To increase the discussion of the paper on the implications and the significance of the findings, some considerations (see below) on the potential applications, along with new references, have been added in the Discussion section (outlined with yellow color).

“It has already been demonstrated that most LED products with high local strain have poor photoelectric performance and decrease in a short time [49], and in addition, the reliability also includes the lifetime mainly [50]. Recently, many articles have mentioned that the most important factor in affecting reliability is the thermal management problem of the LED packaging structure [51, 52]. Therefore, the reliability of LED products depends on the cooling channels with reasonable design and on the materials with high heat conduction performance. Moreover, the local strain in the chip affects the emission characteristics of LEDs, the operational stability is reduced, and the strained interfaces cause an increased defect density. Power conversion efficiency can be strongly affected by local strain. In the current study, the distribution of strain inside the device has been systematically investigated by testing the effect of different assembly parameters and the temperature dependence of the Raman signal. Therefore, the characteristics of the sub-strates and assembly parameters that induce the least stress development have been identified in this work, from the comparison between the various samples.”

  1. The manuscript presents a detailed temperature-dependent strain analysis of the semiconductor materials. While the results at different temperatures provide valuable insights, it is suggested that the authors expand on the implications of these temperature variations for practical applications. How do these temperature-induced strains impact the overall performance and reliability of the semiconductor devices in real-world scenarios? Additionally, a discussion on potential methods to mitigate or manage these temperature-induced strains in practical applications would greatly enhance the applicability of the research findings.

We express our gratitude to the reviewer for the insightful comment. In response, we have incorporated additional information on the temperature aspects of our study in the Introduction and in the Discussion (outlined with yellow color).

“For applications in diverse environments, where they are often subjected to high temperatures and thermal cycling, one critical aspect that influences the widespread use of LEDs is their reliability and resistance, particularly in the face of extreme temperature conditions. Ensuring the reliability of LED devices under such challenging conditions has become a focal point in the pursuit of sustainable and efficient lighting solutions.”

"One of the particularly interesting results to underline is related to the position of the Si and Gan Raman peaks recorded before and after thermal treatments: since it remains unchanged, it indicates reversibility of the process and therefore a good thermal resistance of the samples. This is a good result, since many electronic devices have to operate at high temperature and are exposed to temperature cycles with large temperature difference; thus, they have to withstand strong cyclic thermomechanical stress. Several failure modes are possible up to intense cracking of the solder joints. Further advanced testing is required to verify the formation of defects in welded components and interconnection materials when a large number of thermal cycles occur. As an example, crack propagation in solder joints has already been identified by measuring the relative thermal resistance using transient thermal analysis [53, 54].”

  1. Figure 1 (panels c and d): The microscopic images lack scale bars and size annotations. Adding these details is crucial for understanding the scale of the observed phenomena.

The authors thank the reviewer for this observation. We corrected the figures by adding the missing scales.

  1. Figures 4 and 5: The text in these figures is too small and challenging to read. Consider resizing or redesigning these figures for better readability.

The authors thank the reviewer for this comment. We have corrected the figures, by rescaling them.

FINAL OBSERVATION. The manuscript requires language editing to improve readability and professionalism. The current level of English distracts from the content's quality. I recommend a thorough language revision, preferably by a native English speaker with technical knowledge in this field.

We have improved the level of English with a thorough language review (see parts outlined with a green color).

Reviewer 2 Report

Comments and Suggestions for Authors

This paper reports on the strain state of GaN LEDs in a module via micro Raman. Since the strain can affect the emission characteristics of LEDs, it is crucial to identify the source of strain at all levels. While this work provides some insights into how the integration process induces strain in LEDs, it lacks an explanation of how the strain actually affects the LED performance, which is probably more important information for readers.

- Please provide scales for Fig. 1 c and d.

- If possible, discuss how the local strain in the chip affects the emission characteristics of LEDs and the reliability of the LED module. 

Author Response

We thank the referee very much for the time he/she dedicated to read our paper and for the valuable comments provided.

  1. Please provide scales for Fig. 1 c and d.

The authors thank the reviewer for this observation. We corrected the figures by adding the missing scales.

While this work provides some insights into how the integration process induces strain in LEDs, it lacks an explanation of how the strain actually affects the LED performance, which is probably more important information for readers.

  1. If possible, discuss how the local strain in the chip affects the emission characteristics of LEDs and the reliability of the LED module.

To integrate the paper as suggested, the implications of the strain on LED performances along with new references, have been added in the Discussion section (outlined with yellow color).

“It has already been demonstrated that most LED products with high local strain have poor photoelectric performance and decrease in a short time [49], and in addition, the reliability also includes the lifetime mainly [50]. Recently, many articles have mentioned that the most important factor in affecting reliability is the thermal management problem of the LED packaging structure [51, 52]. Therefore, the reliability of LED products depends on the cooling channels with reasonable design and on the materials with high heat conduction performance. Moreover, the local strain in the chip affects the emission characteristics of LEDs, the operational stability is reduced, and the strained interfaces cause an increased defect density. Power conversion efficiency can be strongly affected by local strain. In the current study, the distribution of strain inside the device has been systematically investigated by testing the effect of different assembly parameters and the temperature dependence of the Raman signal. Therefore, the characteristics of the substrates and assembly parameters that induce the least stress development have been identified in this work, from the comparison between the various samples.”

Round 2

Reviewer 2 Report

Comments and Suggestions for Authors

The authors have adequately addressed my concerns. I have no further comments.